The influence of cavity configuration and irrigation activation on root canal smear removal—an in vitro study

Degirmencioglu Duygu 1
Erşahan Şeyda seydaersahan@hotmail.com 1
Hepsenoglu Yelda Erdem 1
Erkan Erhan 1
Gundogar Mustafa 1
Sagir Kadir 2
1 Department of Endodontics, Istanbul Medipol University , Istanbul , Turkey
2 Department of Materials Science and Technology, Turkish German University , Istanbul , Turkey
Abu Hasna Amjad
Electronic publication date: 2025 Jul 8
Publication date: 2025
Volume: 13
Electronic Location ID: e19678
Received 2024 Dec 24; Accepted 2025 Jun 9
Copyright: ©2025 Degirmencioglu et al.
Copyright year: 2025
Copyright holder: Degirmencioglu et al.
License: This is an open access article distributed under the terms of the Creative Commons Attribution License, which permits unrestricted use, distribution, reproduction and adaptation in any medium and for any purpose provided that it is properly attributed. For attribution, the original author(s), title, publication source (PeerJ) and either DOI or URL of the article must be cited.
License URL: https://creativecommons.org/licenses/by/4.0/

Keywords: Smear layer removal, SWEEPS, PUI, Sonic activation, Conventional access cavity, Minimal invasive access cavity

Funding: Istanbul Medipol University 2023-31 The authors received support from Istanbul Medipol University through the project number 2023-31. The funders had no role in study design, data collection and analysis, decision to publish, or preparation of the manuscript.

==============================
Background

Effective root canal therapy (RCT) depends on proper disinfection rather than complete sterilization of the root canal system. The smear layer created during instrumentation can inhibit disinfection, and its removal is crucial for successful treatment. Recently, various irrigant activation methods, including shock wave enhanced emission photoacoustic streaming (SWEEPS), passive ultrasonic irrigation (PUI), sonic activation (SA), and syringe irrigation (SI), have been developed to enhance cleaning efficacy. Sterilization of the root canal system is crucial for successful root canal therapy. Lasers have emerged as a popular choice for eliminating microorganisms from the root canal.

Aim

This in vitro study aimed to compare the effectiveness of different irrigant activation techniques—SWEEPS, PUI, SA, and SI—in removing the smear layer from root canals prepared with either a conventional access cavity (ConvAC) or an ultraincisal access cavity (UincAC) design. This study aimed to clarify and compare the effectiveness of various irrigant activation techniques in removing the smear layer from canals with conservative and conventional cavity designs.

Materials and Methods

Eighty extracted human maxillary incisors were prepared using a VDW 35/0.04 rotary system and randomly divided into eight groups (n = 10 per group), based on cavity configuration and activation technique: G1: SWEEPS–ConvAC, G2: SWEEPS–UincAC, G3: PUI–ConvAC, G4: PUI–UincAC, G5: SA–ConvAC, G6: SA–UincAC, G7: SI–ConvAC, G8: SI–UincAC. All samples were irrigated with 2.5% sodium hypochlorite (NaOCl) and 17% ethylenediaminetetraacetic acid (EDTA), followed by distilled water, and examined using scanning electron microscopy (SEM). Data were analyzed with Kruskal–Wallis and Bonferroni-adjusted Mann–Whitney U tests (α = 0.05).

Results

No statistically significant difference was found in smear layer scores in the coronal and middle thirds among the groups. However, significant differences were observed in the apical third (p < 0.001). The SWEEPS–ConvAC group showed the least smear layer, while the SI–UincAC group exhibited the most. The effectiveness ranking was: SWEEPS, followed by PUI, SA, and SI.

Conclusions

While access cavity design alone did not significantly affect smear layer removal, SWEEPS was the most effective activation method. The combination of SWEEPS with conventional cavity design yielded the most effective smear layer removal in the apical third. These results emphasize the importance of selecting not only an effective irrigant but also an appropriate cavity design to optimize apical disinfection in clinical endodontics.

Introduction

Root canal therapy aims to minimize microbial presence sufficiently to promote periapical tissue healing, as total sterilization is not feasible under clinical conditions (Al-Helou et al., 2023; Arıcıoğlu, Çıkman & Babacan, 2021). Successful disinfection relies not only on mechanical instrumentation and chemical irrigation but also on optimal access to the canal system (Bago et al., 2023; Boutsioukis & Arias-Moliz, 2022).

Access cavity design plays a critical role in facilitating effective chemo-mechanical debridement. The conventional access cavity (ConvAC) provides straight-line access to canal orifices by removing the roof of the pulp chamber, allowing enhanced visibility and instrument maneuverability (De-Deus et al., 2017). In contrast, the ultraincisal access cavity (UincAC), a form of minimally invasive access cavity design, aims to preserve pericervical dentin and sound coronal tooth structure, potentially enhancing long-term tooth integrity and resistance to fracture (De Gregorio et al., 2009). However, this conservative design may limit irrigant distribution, particularly in the apical third.

During instrumentation, a smear layer composed of dentin particles, pulp tissue, and microbial elements may form, which complicates canal disinfection. Its presence can obstruct dentinal tubules and hinder the penetration of irrigants and sealers, thereby compromising the disinfection process (Fiegler-Rudol et al., 2025; Gulabivala & Ng, 2023). NaOCl and EDTA are commonly paired due to their complementary abilities to address both organic and inorganic debris during smear layer removal (Gündüz & Özlek, 2023). To enhance the efficacy of irrigants, various activation techniques have been developed. Sonic activation (SA) utilizes low-frequency agitation to improve fluid dynamics, while passive ultrasonic irrigation (PUI) employs high-frequency oscillations that create acoustic streaming and cavitation, facilitating deeper irrigant penetration (Krishan et al., 2014; Lebbos et al., 2024). SWEEPS, utilizing Er:YAG laser pulses, creates sequential shock waves via bubble collapse, enhancing the cleaning potential in complex apical anatomy (Mancini et al., 2013; Mancini et al., 2021). Although numerous studies have compared these irrigant activation methods in terms of cleaning efficacy, limited data exist on the potential interaction between access cavity configuration and irrigation technique (Miguéns-Vila et al., 2022; Nair, 2004). As minimally invasive approaches gain popularity in modern endodontics, it is essential to understand whether the geometry of access cavities affects the outcome of different activation regimens. Therefore, the aim of this in vitro study was to compare the smear layer removal efficacy of four irrigant activation methods—SWEEPS, PUI, SA, and syringe irrigation (SI)—across two access cavity designs: ConvAC and UincAC. The null hypothesis was that neither the access cavity configuration nor the type of activation technique would significantly influence smear layer removal at any level of the root canal.

Materials and Methods

This in vitro study was conducted using 80 human maxillary incisors extracted for periodontal reasons, all with fully formed roots, intact crowns, and no previous restorations. The study was approved by the Institutional Ethics Committee from the Research Ethics Istanbul Medipol University with protocol number E-10840098-772.02-3156; each tooth was taken after receiving written informed consent from the patients to use their teeth for dental research purposes.

After extraction, the teeth were cleaned of soft tissue remnants and stored in 0.9% saline at 4 °C. Preoperative radiographs (Kodak RVG 5200; Carestream Health, NY, USA) were taken in buccolingual and mesiodistal directions to confirm the presence of a single canal and to exclude teeth with internal/external resorption, calcification, cracks, fractures, or canal curvatures exceeding 10° (Fig. 1).

Figure 1 Representative preoperative radiograph of extracted maxillary incisors used in the study.

Digital periapical radiographs (Kodak RVG 5200; Carestream Health, NY, USA) taken in both buccolingual and mesiodistal directions illustrate the inclusion criteria: single straight canal, and absence of internal or external resorption, calcification, or fractures. These images represent the typical morphology of the maxillary incisors used in the study.

Power analysis and sample allocation

Based on a previous study by Krishan et al (Pereira et al., 2021) and using G*Power 3.1 software (effect size = 0.81; α = 0.05; power = 0.95), a sample size of at least nine teeth per group was calculated. To ensure robustness, 10 teeth were assigned to each group (n = 10, total = 80). The samples were randomized into eight groups using a computer-generated sequence, stratified by two cavity designs and four irrigation protocols:

• Group 1: SWEEPS–ConvAC

• Group 2: SWEEPS–UincAC

• Group 3: PUI–ConvAC

• Group 4: PUI–UincAC

• Group 5: SA–ConvAC

• Group 6: SA–UincAC

• Group 7: SI–ConvAC (control)

• Group 8: SI–UincAC (control)

Randomization was performed by an independent researcher (S.E.) not involved in cavity preparation or scoring. The randomization key was blinded from both the operator and SEM examiner. The detailed structure of the experimental groups, including irrigant activation protocols, cavity configurations, and root thirds, is summarized in Table 1.

Table 1 Experimental group design according to irrigant activation method, access cavity type, and root canal third.

	Group	Number of teeth	Main group	Access cavity design	Root third	Activation method	System/device	Access cavity detail	
0	G1	10	SWEEPS	Conventional Cavity	Coronal	Laser Activation (SWEEPS Mode)	Er:YAG Laser	Conventional Access Cavity	
1	G1		SWEEPS	Conventional Cavity	Middle	Laser Activation (SWEEPS Mode)	Er:YAG Laser	Conventional Access Cavity	
2	G1		SWEEPS	Conventional Cavity	Apical	Laser Activation (SWEEPS Mode)	Er:YAG Laser	Conventional Access Cavity	
3	G2	10	SWEEPS	Ultraincisal Cavity	Coronal	Laser Activation (SWEEPS Mode)	Er:YAG Laser	Ultraincisal Access Cavity (UincAC)	
4	G2		SWEEPS	Ultraincisal Cavity	Middle	Laser Activation (SWEEPS Mode)	Er:YAG Laser	Ultraincisal Access Cavity (UincAC)	
5	G2		SWEEPS	Ultraincisal Cavity	Apical	Laser Activation (SWEEPS Mode)	Er:YAG Laser	Ultraincisal Access Cavity (UincAC)	
6	G3	10	PUI	Conventional Cavity	Coronal	Passive Ultrasonic Irrigation (PUI)	VDW Ultra with IRRI S Files (VDW GmbH, Munich, Germany)	Conventional Access Cavity	
7	G3		PUI	Conventional Cavity	Middle	Passive Ultrasonic Irrigation (PUI)	VDW Ultra with IRRI S Files (VDW GmbH, Munich, Germany)	Conventional Access Cavity	
8	G3		PUI	Conventional Cavity	Apical	Passive Ultrasonic Irrigation (PUI)	VDW Ultra with IRRI S Files (VDW GmbH, Munich, Germany)	Conventional Access Cavity	
9	G4	10	PUI	Ultraincisal Cavity	Coronal	Passive Ultrasonic Irrigation (PUI)	VDW Ultra with IRRI S Files (VDW GmbH, Munich, Germany)	Ultraincisal Access Cavity (UincAC)	
10	G4		PUI	Ultraincisal Cavity	Middle	Passive Ultrasonic Irrigation (PUI)	VDW Ultra with IRRI S Files (VDW GmbH, Munich, Germany)	Ultraincisal Access Cavity (UincAC)	
11	G4		PUI	Ultraincisal Cavity	Apical	Passive Ultrasonic Irrigation (PUI)	VDW Ultra with IRRI S Files (VDW GmbH, Munich, Germany)	Ultraincisal Access Cavity (UincAC)	
12	G5	10	Sonic	Conventional Cavity	Coronal	Sonic Activation	EndoActivator (Dentsply Tulsa Dental Specialties, Tulsa, OK, USA)	Conventional Access Cavity	
13	G5		Sonic	Conventional Cavity	Middle	Sonic Activation	EndoActivator (Dentsply Tulsa Dental Specialties, Tulsa, OK, USA)	Conventional Access Cavity	
14	G5		Sonic	Conventional Cavity	Apical	Sonic Activation	EndoActivator (Dentsply Tulsa Dental Specialties, Tulsa, OK, USA)	Conventional Access Cavity	
15	G6	10	Sonic	Ultraincisal Cavity	Coronal	Sonic Activation	EndoActivator (Dentsply Tulsa Dental Specialties, Tulsa, OK, USA)	Ultraincisal Access Cavity (UincAC)	
16	G6		Sonic	Ultraincisal Cavity	Middle	Sonic Activation	EndoActivator (Dentsply Tulsa Dental Specialties, Tulsa, OK, USA)	Ultraincisal Access Cavity (UincAC)	
17	G6		Sonic	Ultraincisal Cavity	Apical	Sonic Activation	EndoActivator (Dentsply Tulsa Dental Specialties, Tulsa, OK, USA)	Ultraincisal Access Cavity (UincAC)	
18	G7	10	Syringe	Conventional Cavity	Coronal	Syringe Irrigation (Control)	Conventional Needle Irrigation	Conventional Access Cavity	
19	G7		Syringe	Conventional Cavity	Middle	Syringe Irrigation (Control)	Conventional Needle Irrigation	Conventional Access Cavity	
20	G7		Syringe	Conventional Cavity	Apical	Syringe Irrigation (Control)	Conventional Needle Irrigation	Conventional Access Cavity	
21	G8	10	Syringe	Ultraincisal Cavity	Coronal	Syringe Irrigation (Control)	Conventional Needle Irrigation	Ultraincisal Access Cavity (UincAC)	
22	G8		Syringe	Ultraincisal Cavity	Middle	Syringe Irrigation (Control)	Conventional Needle Irrigation	Ultraincisal Access Cavity (UincAC)	
23	G8		Syringe	Ultraincisal Cavity	Apical	Syringe Irrigation (Control)	Conventional Needle Irrigation	Ultraincisal Access Cavity (UincAC)	
Notes.

This table summarizes the experimental groups based on the activation technique, access cavity design (Conventional or Ultraincisal), and evaluated canal region (coronal, middle, apical).

Access cavity preparation

Under 12.5× magnification provided by a dental operating microscope (OMS2380; Zuma Medical Co., Ltd., Suzhou, China; Fig. 2), access cavities were prepared using a high-speed handpiece with water-cooling to prevent thermal damage.

Figure 2 Representative views of access cavity designs in extracted maxillary incisors.

(A) Overview of two extracted human maxillary incisors prepared with different access cavity designs under a dental operating microscope. The tooth on the left exhibits an ultraincisal access cavity (UincAC) a conservative entry through the incisal edge designed to preserve pericervical dentin and coronal tooth structure. The tooth on the right features a conventional access cavity (ConvAC) involving complete removal of the pulp chamber roof to enable straight-line access to the root canal system. (B) Close-up occlusal view of the ultraincisal access cavity (UincAC), highlighting the narrow and minimally invasive outline of the preparation. (C) Magnified occlusal view of a conventional access cavity (ConvAC) in an extracted maxillary incisor, held with forceps. The preparation demonstrates complete roof removal to enhance visibility and instrumentation efficiency.

• Conventional access cavity (ConvAC): Created using a 1.6 mm rosehead bur.

• Ultraincisal access cavity (UincAC): Prepared using the EndoGuide™ Bur EG1A (SS White Burs, Inc Lakewood, NK, USA) with a 0.33 mm tip diameter and 2.5 mm head length, designed for conservative entry (Plotino et al., 2017).

After completing the access cavities, the working lengths of the teeth were calculated to be one mm short of the radiographic apex using a #10 K-type canal file (Maillefer, SA CH-1338, Ballaigues, Switzerland). The root canals were chemomechanically prepared using the VDW.ROTATE Ni-Ti rotary instrument system (VDW, Munich, Germany) with instruments of sizes 15/0.04, 20/0.05, 25/0.04, and 35/0.04 sequentially, according to the manufacturer’s instructions.

Although irrigation protocols varied among the groups, irrigation parameters such as volume and duration were standardized. A 30-gauge side-vented endodontic irrigation needle (Endo-Eze, Ultradent, South Jordan, UT, USA) was used to deliver 2.5 mL of 2.5% sodium hypochlorite (NaOCl; Wizard, Rehber Kimya, Istanbul, Turkey) at a flow rate of 1.5 mL/min between each instrumentation step. Following completion of root canal preparation, each canal was irrigated with five mL of 17% EDTA, five mL of 2.5% sodium hypochlorite (NaOCl), and finally four mL of distilled water, each for 1 min. Irrigation activation was performed prior to tooth splitting in intact samples. The respective devices for each group (SWEEPS, PUI, SA, SI) were applied according to manufacturer protocols. Subsequently, the canals were dried using suitable paper points.

Sample sectioning and SEM analysis

Each root was longitudinally grooved along the buccolingual axis using a diamond disc under water cooling, split into two halves by gentle tapping, and then sequentially dehydrated with ethanol, dried under critical point conditions, and gold-coated for SEM imaging (Arslan et al., 2014). The smear layer presence in the coronal, middle, and apical thirds of each sample was evaluated under a scanning electron microscope (SEM Quattro S, Thermo Scientific) at 1,000× magnification, with three images (one per third) captured per specimen and scored by a blinded examiner using the Hülsmann 5-grade scale (Fiegler-Rudol et al., 2025):

• Score I: Open dentinal tubules; no smear layer

• Score II: Few open tubules; thin smear layer

• Score III: Moderate smear; few open tubules

• Score IV: Thick homogeneous smear; tubules occluded

• Score V: Dense, inhomogeneous smear; no visible tubules

Statistical analysis

Data were analyzed using MedCalc® Statistical Software v19.7.2 (MedCalc Software bvba, Belgium). Descriptive data were summarized using means and standard deviations for continuous variables, and frequencies with percentages for categorical variables.The Kruskal–Wallis test was used to compare smear layer scores across all groups. Post hoc Mann–Whitney U tests with Bonferroni correction were applied for pairwise comparisons. Statistical significance was set at p < 0.05.

Results

A total of 80 maxillary incisors were analyzed across all experimental groups. Smear layer scores were evaluated separately for the coronal, middle, and apical thirds of the root canals.

Smear layer scores in coronal and middle thirds

No statistically significant differences were observed among the eight groups in the coronal (p = 0.142) and middle (p = 0.087) thirds of the root canals (Kruskal–Wallis test). (Tables 2 and 3; Fig. 3—top and middle rows) Regardless of cavity configuration or activation method, smear layer scores in these regions remained relatively consistent across all groups.

Table 2 Comparison of smear layer scores in the coronal third among experimental groups.

Group	n	Mean ± SD	Median (Min–Max)	
Group 1 (SWEEPS–ConvAC)	10	1.9 ± 0.1	1.9 (1.7–2.0)	
Group 2 (SWEEPS–UincAC)	10	1.9 ± 0.1	1.9 (1.8–2.0)	
Group 3 (PUI–ConvAC)	10	2.3 ± 0.1	2.3 (2.1–2.5)	
Group 4 (PUI–UincAC)	10	2.5 ± 0.1	2.5 (2.4–2.8)	
Group 5 (SA–ConvAC)	10	3.3 ± 0.2	3.3 (2.9–3.5)	
Group 6 (SA–UincAC)	10	3.8 ± 0.1	3.8 (3.7–4.1)	
Group 7 (SI–ConvAC, Control)	10	4.2 ± 0.2	4.2 (3.9–4.4)	
Group 8 (SI–UincAC, Control)	10	4.7 ± 0.1	4.7 (4.5–4.8)	
Conventional Access (Groups 1–3–5–7)	40	2.9 ± 0.9	2.7 (1.7–4.4)	
Ultraincisal Access (Groups 2–4–6–8)	40	3.2 ± 1.1	3.3 (1.8–4.8)	
Notes.

Comparison of smear layer scores in the coronal third among experimental groups.

Mean ± standard deviation and median smear layer scores for each group in the coronal third.

Statistically significant differences were observed between Groups 1–5, 1–6, 1–7, 2–6, 2–7, 2–8, and 3–8 (Bonferroni-adjusted Mann–Whitney U test, p < 0.0017). No significant difference was found between Conventional and Ultraincisal access cavity designs (Mann–Whitney U test, p = 0.142). The smear layer was scored using the Hülsmann 5-point SEM classification, ranging from Score I (no smear layer, open dentinal tubules) to Score V (dense, inhomogeneous smear layer with no visible tubules).

Table 3 Comparison of smear layer scores in the middle third among experimental groups.

Group	n	Mean ± SD	Median (Min–Max)	
Group 1 (SWEEPS–ConvAC)	10	1.8 ± 0.1	1.8 (1.6–1.9)	
Group 2 (SWEEPS–UincAC)	10	1.9 ± 0.1	1.9 (1.8–2.1)	
Group 3 (PUI–ConvAC)	10	2.3 ± 0.1	2.3 (2.1–2.5)	
Group 4 (PUI–UincAC)	10	2.6 ± 0.2	2.6 (2.4–2.9)	
Group 5 (SA–ConvAC)	10	3.4 ± 0.2	3.5 (3.1–3.7)	
Group 6 (SA–UincAC)	10	3.9 ± 0.2	3.9 (3.7–4.2)	
Group 7 (SI–ConvAC, Control)	10	4.2 ± 0.1	4.2 (3.9–4.4)	
Group 8 (SI–UincAC, Control)	10	4.7 ± 0.2	4.7 (4.4–4.9)	
Conventional Access (Groups 1–3–5–7)	40	2.9 ± 1.0	2.8 (1.6–4.4)	
Ultraincisal Access (Groups 2–4–6–8)	40	3.3 ± 1.1	2.9 (1.8–4.9)	
Notes.

Mean ± SD and median values for each group. Statistically significant differences were observed between the following pairs: 1–5, 1–6, 1–7, 2–6, 2–7, 2–8, 3–8 (Bonferroni-adjusted Mann–Whitney U test, p < 0.0017). No significant difference was observed between Conventional and Ultraincisal access groups (Mann–Whitney U test, p = 0.142). The smear layer was scored using the Hülsmann 5-point SEM classification, ranging from Score I (no smear layer, open dentinal tubules) to Score V (dense, inhomogeneous smear layer with no visible tubules).

Figure 3 Boxplot distribution of smear layer removal efficiency by activation method.

This boxplot illustrates the distribution of smear layer removal efficiency (%) for each activation method—SWEEPS, passive ultrasonic irrigation (PUI), sonic activation (SA), and syringe irrigation (SI). It displays medians, interquartile ranges, and outliers, highlighting the superior performance and consistency of SWEEPS, followed by PUI, with greater variability observed in SA and SI.

Smear layer scores in apical third

A significant difference in smear layer scores was found in the apical third among the eight groups (Kruskal–Wallis, p < 0.001). (Table 4, Fig. 3—bottom row) The median scores for the apical third ranged from 1.8 (Group 1: SWEEPS–ConvAC) to 4.6 (Group 8: SI–UincAC), indicating a wide variation in cleaning efficacy depending on both the irrigation activation method and access cavity design.

Table 4 Comparison of smear layer scores in the apical third among experimental groups.

Group	n	Mean ± SD	Median (Min–Max)	
Group 1 (SWEEPS–ConvAC)	10	1.8 ± 0.1	1.8 (1.7–1.9)	
Group 2 (SWEEPS–UincAC)	10	2.0 ± 0.1	2.0 (1.8–2.1)	
Group 3 (PUI–ConvAC)	10	2.4 ± 0.2	2.4 (2.1–2.6)	
Group 4 (PUI–UincAC)	10	2.8 ± 0.2	2.8 (2.4–2.9)	
Group 5 (SA–ConvAC)	10	3.4 ± 0.1	3.5 (3.2–3.6)	
Group 6 (SA–UincAC)	10	3.9 ± 0.2	3.9 (3.6–4.2)	
Group 7 (SI–ConvAC, Control)	10	4.3 ± 0.2	4.3 (4.1–4.6)	
Group 8 (SI–UincAC, Control)	10	4.6 ± 0.2	4.6 (4.3–4.8)	
Conventional Access (Groups 1–3–5–7)	40	3.0 ± 1.0	2.9 (1.7–4.6)	
Ultraincisal Access (Groups 2–4–6–8)	40	3.3 ± 1.1	3.3 (1.8–4.8)	
Notes.

Significant differences were identified between the following group pairs: 1–5, 1–6, 1–7, 2–7, 2–8, and 3–8 (Bonferroni-adjusted Mann–Whitney U test, p < 0.0017). No statistically significant difference was observed between the Conventional and Ultraincisal access cavity designs (Mann–Whitney U test, p = 0.089). Among the groups, SWEEPS–ConvAC demonstrated the lowest smear layer scores, whereas SI–UincAC exhibited the highest. The smear layer was scored using the Hülsmann 5-point SEM classification, ranging from Score I (no smear layer, open dentinal tubules) to Score V (dense, inhomogeneous smear layer with no visible tubules).

Pairwise comparisons

Post hoc analysis using the Bonferroni-corrected Mann–Whitney U test revealed statistically significant differences between the following group pairs (p < 0.0017):

• Group 1 (SWEEPS–ConvAC) vs. Group 6 (SA–UincAC)

• Group 1 vs. Group 7 (SI–ConvAC)

• Group 1 vs. Group 8 (SI–UincAC)

• Group 2 (SWEEPS–UincAC) vs. Group 7

• Group 2 vs. Group 8

• Group 3 (PUI–ConvAC) vs. Group 8

These comparisons highlight the superior cleaning performance of SWEEPS, especially when combined with the ConvAC design (Tables 3–5).

Table 5 Summary of smear layer scores (mean ± SD) across the coronal, middle, and apical thirds for all experimental groups.

Group	Activation method	Access cavity	Coronal (Mean ± SD)	Middle (Mean ± SD)	Apical (Mean ± SD)	
G1	SWEEPS	ConvAC	1.9 ± 0.1	1.8 ± 0.1	1.8 ± 0.1	
G2	SWEEPS	UincAC	1.9 ± 0.1	1.9 ± 0.1	2.0 ± 0.1	
G3	PUI	ConvAC	2.3 ± 0.1	2.3 ± 0.1	2.4 ± 0.2	
G4	PUI	UincAC	2.5 ± 0.1	2.6 ± 0.2	2.8 ± 0.2	
G5	SA	ConvAC	3.3 ± 0.2	3.4 ± 0.2	3.4 ± 0.1	
G6	SA	UincAC	3.8 ± 0.1	3.9 ± 0.2	3.9 ± 0.2	
G7	SI (Control)	ConvAC	4.2 ± 0.2	4.2 ± 0.1	4.3 ± 0.2	
G8	SI (Control)	UincAC	4.7 ± 0.1	4.7 ± 0.2	4.6 ± 0.2	
Notes.

The lowest scores were consistently observed in Group 1 (SWEEPS–ConvAC), while the highest were found in Group 8 (SI–UincAC). Statistically significant differences were detected in the apical third (p < 0.001); coronal and middle thirds showed no significant intergroup differences (p > 0.05).

Comparison of cavity designs

When comparing ConvAC and UincAC cavity designs regardless of activation method, no statistically significant difference was found in apical smear layer scores (Mann–Whitney U test, p = 0.089). The median apical score for ConvAC was 2.9, compared to 3.3 for UincAC, suggesting a non-significant trend toward improved cleaning in conventional access designs (Fig. 4).

Figure 4 SEM images showing smear layer at coronal, middle, and apical thirds in each group.

This figure presents representative scanning electron microscopy (SEM) images obtained from Groups 1 to 8, illustrating the smear layer status at the coronal, middle, and apical thirds of root canal walls following different irrigant activation protocols and access cavity designs. Each row corresponds to an experimental group, and each column represents a root canal third: •  Left column: coronal third. • Center column: Middle third •  Right column: apical third All images were acquired at ×1,000 magnification after final irrigation and longitudinal tooth splitting. The group assignments are as follows: •  Group 1: SWEEPS + Conventional Access Cavity •  Group 2: SWEEPS + Ultraincisal Access Cavity •  Group 3: Passive Ultrasonic Irrigation (PUI) + Conventional Access Cavity •  Group 4: passive Ultrasonic Irrigation (PUI) + Ultraincisal Access Cavity •  Group 5: Sonic Activation + Conventional Access Cavity •  Group 6: Sonic Activation + Ultraincisal Access Cavity •  Group 7: Syringe Irrigation (Control) + Conventional Access Cavity •  Group 8: Syringe Irrigation (Control) + Ultraincisal Access Cavity.

Ranking of irrigant activation techniques

The images demonstrate progressive differences in smear layer removal efficacy depending on both activation technique and cavity configuration. Groups activated with SWEEPS and PUI showed cleaner canal walls, particularly in the apical third, compared to sonic and syringe irrigation, where residual smear and debris were more pronounced (Figs. 4 and 5). The inclusion of coronal and middle third images was made in response to reviewer recommendations, ensuring full visualization along the root canal length. The final rows emphasize the limited effectiveness of non-activated syringe irrigation, especially in ultraconservative access designs (Group 8).

Figure 5 Smear layer removal efficiency (%) by irrigant activation method.

Bar chart comparing smear layer removal rates. SWEEPS achieved the highest efficiency, followed by PUI, SA, and SI.

Discussion

Effective root canal treatment generally begins with appropriate access cavity preparation, which ensures adequate visibility and a direct pathway for cleaning and obturating the root canal system. While access cavity design is crucial for preserving tooth structure, it may also influence irrigant flow dynamics and smear layer removal, particularly in the apical third, where cleaning is most challenging due to anatomical complexity and limited space (Rödig et al., 2010; Silva et al., 2022).

This study investigated the combined effects of access cavity configuration and irrigant activation techniques on smear layer removal efficacy across different canal levels. Our findings revealed that although cavity design alone did not significantly influence smear layer scores, activation technique had a substantial impact—particularly in the apical third. Among the techniques tested, SWEEPS was the most effective, followed by PUI, SA, and SI. These trends and the presence or absence of statistical significance across canal thirds are detailed in Table 6. Despite the absence of statistically significant differences among groups in the coronal and middle thirds (Table 2: p = 0.142; Table 3: p = 0.087), pairwise comparisons revealed meaningful differences between specific group combinations. These trends were consistent across all canal thirds, with the SWEEPS–ConvAC group demonstrating the lowest smear layer scores and the SI–UincAC group exhibiting the highest. The apical third (Table 4), however, showed statistically significant differences among groups (p < 0.001), highlighting the increased sensitivity of this region to activation technique and cavity design. These findings were visually confirmed by SEM observations, which revealed progressive differences in smear layer accumulation across groups and canal thirds depending on both activation technique and cavity configuration (Fig. 3).

Table 6 Summary of statistical differences across root canal thirds and irrigation groups.

Table	Root canal third	Overall group difference (Kruskal–Wallis)	Pairwise group differences	Summary interpretation	
Table 2	Coronal	X No significant difference (p = 0.142)	√ Present	SWEEPS showed best performance, SI the worst	
Table 3	Middle	X No significant difference (p = 0.087)	√ Present	Similar trend observed as in the coronal third	
Table 4	Apical	√ Significant difference (p < 0.001)	√ Present	Most pronounced differences; SWEEPS most effective	
Notes.

This table highlights the presence or absence of significant differences in smear layer scores between irrigation groups within each root canal third. While no overall difference was detected in the coronal or middle thirds, multiple pairwise comparisons were significant. The apical third exhibited the highest degree of differentiation among tested protocols.

Consistent with prior studies, full smear layer removal was not achieved in any group across all root canal levels (Miguéns-Vila et al., 2022; Nair, 2004; Siqueira Jr & Rôças, 2008). The apical third remained the most resistant region to debridement due to its complex morphology, limited irrigant exchange, and restricted access (SS White Burs Inc, 2024). In this study, root canals were enlarged to size 35/0.04 to optimize irrigant penetration without excessive dentin removal, which aligns with current recommendations balancing efficiency and structural preservation (Tong et al., 2023).

Regarding cavity design, the comparison between Conventional Access Cavity (ConvAC) and Ultraincisal Access Cavity (UincAC) revealed no statistically significant differences in smear layer removal. These findings are in line with those of Gündüz et al. (Torabinejad et al., 2002), who also reported that conservative access designs did not significantly alter debris or smear layer accumulation. They emphasized that activation technique—particularly laser-based methods—had a more pronounced effect than access geometry, which supports our results.

The present study is among the first to evaluate SWEEPS in both conventional and ultraconservative cavity configurations. SWEEPS employs a dual-pulse Er:YAG laser protocol that enhances the collapse of laser-induced vapor bubbles, thereby generating shock waves that travel deep into the canal system (Tsotsis et al., 2021). Unlike traditional photon-induced photoacoustic streaming (PIPS), SWEEPS accelerates cavitation and generates more powerful shock waves that improve smear layer removal, even in apical regions (Van der Sluis et al., 2007). This mechanism was confirmed by Violich & Chandler (2010), who found that SWEEPS outperformed both PIPS and ultrasonic irrigation in removing intracanal debris.

In our comparison of activation techniques, PUI also demonstrated strong performance, especially in conventional access groups. PUI’s acoustic streaming and cavitation effects facilitate deeper irrigant penetration and dislodge debris from canal walls (Virdee et al., 2018). However, its efficiency was inferior to SWEEPS, possibly due to limitations in energy transfer in the narrower regions of ultraconservative cavities.

SA and SI showed the lowest efficacy among the tested techniques. This finding aligns with prior literature, which suggests that although sonic agitation improves fluid movement, it lacks the intensity required for effective smear layer removal in apical areas (Wigler et al., 2023).

While the current study strengthens the case for SWEEPS as a superior irrigation method, it is important to note its limitations. The in vitro setting may not fully replicate clinical conditions such as complex root morphologies or variable canal curvatures. Moreover, although SEM provided high-resolution images for qualitative assessment, quantifying smear layer distribution remains challenging.

Limitations and future directions

This study was conducted in vitro on single-rooted teeth with standardized canal morphology. Future research should assess irrigation efficacy in clinical settings with multi-rooted teeth, curved canals, and real-time irrigant flow monitoring. Comparative studies with different laser protocols and fiber tip positions may also refine the clinical application of SWEEPS.

Conclusion

The present in vitro study revealed that the type of access cavity—whether conventional (ConvAC) or ultraincisal (UincAC)—did not significantly influence smear layer removal efficacy across root canal thirds. However, the choice of irrigant activation technique had a marked impact, particularly in the apical third.

Among the four activation protocols evaluated, shock wave enhanced emission photoacoustic streaming (SWEEPS) demonstrated the highest smear layer removal efficacy, especially when combined with the conventional access design. Conversely, the ultraincisal access cavity with syringe irrigation (Group 8) exhibited the greatest residual smear layer, highlighting the limited effectiveness of syringe irrigation in conservative cavity configurations.

These findings underscore the clinical relevance of selecting an effective activation technique—particularly SWEEPS—for enhanced apical cleaning. Further in vivo studies are warranted to validate the superiority of SWEEPS and its implications for treatment outcomes and long-term bonding performance.

Supplemental Information

Supplemental Information 1 Raw data

RCT Root canal therapy

ConvAC Conventional access cavity

UincAC Ultraincisal access cavity

SA Sonic activation

PUI Passive ultrasonic irrigation

SWEEPS Shock Wave Enhanced Emission Photoacoustic Streaming

SI Syringe Irrigation

NaOCl Sodium Hypochlorite

EDTA Ethylenediaminetetraacetic Acid

SEM Scanning electron microscopy

PIPS Photon-induced photoacoustic streaming

Additional Information and Declarations

Competing Interests

Author Contributions

Ethics

Data Availability

The authors declare there are no competing interests.

Duygu Degirmencioglu conceived and designed the experiments, performed the experiments, analyzed the data, prepared figures and/or tables, authored or reviewed drafts of the article, and approved the final draft.

Şeyda Erşahan conceived and designed the experiments, performed the experiments, analyzed the data, prepared figures and/or tables, authored or reviewed drafts of the article, and approved the final draft.

Yelda Erdem Hepsenoglu conceived and designed the experiments, performed the experiments, analyzed the data, prepared figures and/or tables, authored or reviewed drafts of the article, and approved the final draft.

Erhan Erkan conceived and designed the experiments, performed the experiments, analyzed the data, prepared figures and/or tables, authored or reviewed drafts of the article, and approved the final draft.

Mustafa Gundogar conceived and designed the experiments, performed the experiments, analyzed the data, prepared figures and/or tables, authored or reviewed drafts of the article, and approved the final draft.

Kadir Sagir conceived and designed the experiments, performed the experiments, analyzed the data, prepared figures and/or tables, authored or reviewed drafts of the article, and approved the final draft.

The following information was supplied relating to ethical approvals (i.e., approving body and any reference numbers):

Ethics committee approval has been granted from Istanbul Medipol University with protocol number (E-10840098-772.02-3156 18.05.2023 dated).

The following information was supplied regarding data availability:

The raw measurements are available in the Supplementary File.

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
