# Peer review of "The influence of cavity configuration and irrigation activation on root canal smear removal—an in vitro study"

_PeerJ, doi:10.7717/peerj.19678_

## Round 0.1 · original submission · Major Revisions

Dear authors,

Thank you for your submission. After a thorough review, we have determined that major revisions are required before further consideration. The manuscript is well written, but it lacks essential details regarding the influence of cavity design on irrigation efficacy, focusing only on the apical region while omitting data from the coronal and middle thirds. Additional SEM images and raw data from these areas are necessary. The methodology requires clarification, particularly in sample preparation and irrigation activation, and the rationale for comparing apical smear layer removal between cavity designs is unclear. The research question needs refinement to align with the study’s objective, and referencing must follow the Vancouver system. Additionally, the data provided are insufficient to assess the impact of cavity design on irrigation activation efficacy.

[**Staff Note; Authors may use any referencing system during submission as long as it is complete and consistent. References will be converted to PeerJ style during production.**]

Reviewer 1 ·

Basic reporting

I have highlighted the comments in the file of the article.

Experimental design

The article is well written but the results emphasise on the apical region without any mention about the middle and coronal areas.

Validity of the findings

Results have to address all the areas examined.

Annotated reviews are not available for download in order to protect the identity of reviewers who chose to remain anonymous.

Reviewer 2 ·

Basic reporting

Abstract
- Background: The distinction between sterilization and disinfection needs clarification. Is sterilization the goal of root canal therapy? The background should be rewritten, as it does not clearly relate to the aim or title of this study.
- Aim: The phrase "canals with conservative and conventional cavity designs" is ambiguous—please clarify its meaning.
- Materials and Methods: How was SEM used to evaluate smear layer removal? The specific methodology should be clearly described.
- Results: Is it p < 0.0017 or p = 0.0017? Ensure consistency and accuracy in statistical reporting.
- Conclusion: This section should be rewritten rather than simply copying the results. Use two to three sentences to summarize the findings, particularly regarding the influence of cavity configuration and irrigation activation on root canal smear layer removal.

Introduction
This section requires extensive revision for logical flow and clarity.

- Paragraph 1:
The statement "Root canal therapy (RCT) aims to eliminate debris and the smear layer" is incorrect. The primary goal of RCT is disinfection or prevention of infection in the root canal system, while debris and smear layer removal are means to achieve this goal.
The logical flow of this paragraph needs significant improvement.

- Paragraph 2:
The reference sequence throughout the manuscript is incorrect and must be properly organized.
The first sentence is not supported by sufficient references—please provide adequate citations.

- Paragraph 3:
The statement "Although previous studies have evaluated different irrigant activation techniques in terms of smear layer removal (ref yaz)" is unclear.
What is "ref yaz"? The reference must be properly formatted.

Experimental design

- The root canals were chemomechanically prepared using the VDW.ROTATE Ni-Ti rotary instrument system (VDW, Munich, Germany) with instruments of sizes 15/0.04, 20/0.05, 25/0.04, and 35/0.04 sequentially, following the manufacturer's instructions. However, size 30 is missing—please clarify.
- How was it ensured that no samples were discarded during preparation for SEM analysis?
- There is no reference provided for the Hülsmann 5-grade scoring system—please include a citation.
- The SEM scoring method lacks detail. How many SEM images were obtained per sample? What magnifications or scale bars were used for SEM imaging?
- The sample size calculation section should be moved to the beginning of the Materials and Methods section for better organization.
- There are two separate "Statistical Analysis" sections, which should be merged into one for clarity.

Validity of the findings

- Tables 1, 2, 3, and 4 should be merged into a single table for clarity and conciseness.
- Figure 1: The SEM image quality is poor and needs improvement. Additionally, a legend should be included to specify which images correspond to each group.
- Figure 2: The meaning of "33" and "41" in the figure needs to be clarified.
- A flowchart illustrating the experimental process should be added to enhance reader comprehension.
- Representative images of different access cavity designs should be included for better visualization.

Additional comments

No.

·

Basic reporting

• The basic language format used throughout the manuscript is quite simple, professional, and very clear to understand.
• No clinical pictures or photos are provided in the manuscript to understand about the various cavity designs.
• No where in the manuscript has it been mentioned about the influence of various cavity designs on the effect of irrigation activation regimens on smear layer removal.
• Only root canal portion has been discussed regarding the smear layer removal and not about the coronal part of pulp chamber.
• Apical measurements used for comparison of Conventional access cavity with Conservative type which is a wrong parameter to assess. We need to study the coronal portion of cavity designs also for assessing the different irrigation regimens efficacy.
• Separate SEM images also needed to be mentioned related to smear layer removal in the coronal part of cavity and not just the apical third.
• In the methodology section, if the samples were sliced in the bucco- lingual direction, then how the irrigating solution was contained inside the pulp chamber and activated. So,the methodology section needs to be corrected.
• Referencing is not according to Vancouver system of indexing.
• Figures mentioned are of high quality.
• Raw data needs to be added related to the smear layer removal from the coronal part of cavity .

Experimental design

• Original research is within the scope of the journal.
• Methodology is not accurate with insufficient information related to the smear layer removal from the pulp chamber portion in regards to different cavity designs types using various irrigation regimens.
• Research question is not well defined and the research needs to be revised to fill up an identified knowledge gap.

Validity of the findings

• Underlying data provided are insufficient to assess the smear layer removal using various irrigation regimens.
• Effect of various cavity designs on the irrigation activation efficacy is not assessed which is a very important parameter.

---

## Round 0.2 · Minor Revisions

Dear authors,
The manuscript is generally clear, with acceptable English and a good writing structure, supported by relevant references. However, minor revisions are required to address key issues in the experimental design and the validity of the findings. The authors should carefully revise the methodology, ensuring that all aspects of the design are clearly explained and any shortcomings identified in the attached comments are corrected. Additionally, it is essential that the data for all root canal thirds—coronal, middle, and apical—are fully presented and discussed to support the study's conclusions.

Reviewer 1 ·

Basic reporting

The paper is clear and English language is acceptable.
It has many related references.
The writing structure is good.

Experimental design

The writer should answer the questions about the design of the method and defects in the results. They are present in the comments in the main file attached

Validity of the findings

All data of the root canal thirds should be stated.

Annotated reviews are not available for download in order to protect the identity of reviewers who chose to remain anonymous.

Reviewer 2 ·

Basic reporting

The authors have well addressed my concerns. No further comments.

Experimental design

The authors have well addressed my concerns. No further comments.

Validity of the findings

The authors have well addressed my concerns. No further comments.

·

Basic reporting

• The basic language format used in the manuscript throughout is quite simple, professional and very clear to understand.
• In the materials and methods section, slight english spelling mistake is there in the term airoter in the access cavity preparation methodology part which needs correction.
• The introduction and background show enough context speaking about advantages of doing a conventional access cavity preparation over ultraincisal access cavity preparation. Introduction to the various irrigation activation systems has been nicely described.
• The article is structured in such a good manner conforming to the esteemed Peerj journal standards.
• Overall the study has been nicely done with all the necessary raw data provided in the manuscript.
• Figures are relevant, high quality, well described and the SEM images are also good describing about the smear layer removal aspect from the coronal, middle and apical third aspect.
• Limitations of this study has also been clearly mentioned by the authors and researchers of this study which shows that further studies in the future can be done in simulated in vivo clinical situations for better evidence based research. Need for inclusion of multi rooted teeth with complex anatomical variations of root canal has also been mentioned as a limiting factor of the study.
• The reference system is according to the Vancover system of referencing.

Experimental design

• Original primary research is within the scope of the journal.
• Methodology is accurate with relevant information provided in regards to the smear layer removal from the pulp and root canal chamber portion in regards to different cavity designs types using various irrigation activation regimens.
• Research question is meaningful, well defined & relevant. It also specifies how the research fills an identified knowledge gap.
• The study has been nicely conducted on extracted teeth using all the necessary parameters, studying about the smear layer removal from coronal, middle and apical third of root canal.
• Rigorous investigation has been performed in an ethical standard way using high techniques.
• Methodology and results sections in the manuscript give sufficient detailed information in the form of box plots distribution of smear layer removal efficiency, SEM images of different groups in coronal, middle and apical third of the root canal, bar diagrams of smear layer removal efficiency % by different activation methods, pictures of the different types of access cavity designs also mentioned.
• Statistical analysis of comparison between smear layer scores between different groups in the coronal, middle and apical third of root canals has been calculated and analysed properly in the study.

Validity of the findings

• All the underlying data have been provided in the manuscript which is sufficient to assess the smear layer removal from different regions of root canal system using various irrigation activation regimens.
• Conclusion of the manuscript provided is well stated and gives a good link to the original research question and supporting results seen which are evidence based.
• The manuscript is now suitable for publication in the esteemed PeerJ journal.

---

## Round 0.3 · accepted · Accept

The authors have satisfactorily addressed all of the reviewers’ comments and made the necessary revisions to improve the manuscript. Although the original reviewers did not re-evaluate the submission, I have carefully evaluated the revised version myself and find that the authors have responded appropriately to all feedback. The current version is clear, scientifically sound, and suitable for publication in its present form.